# CheXagent: Towards a Foundation Model for Chest X-Ray Interpretation

**Zhihong Chen**[1*], **Maya Varma**[1*], **Jean-Benoit Delbrouck**[1*], **Magdalini Paschali**[1]

**Louis Blankemeier**[1], **Dave Van Veen**[1], **Jeya Maria Jose Valanarasu**[1], **Alaa Youssef**[1]

**Joseph Paul Cohen**[1], **Eduardo Pontes Reis**[1], **Emily B. Tsai**[1], **Andrew Johnston**[1]

**Cameron Olsen**[1], **Tanishq Mathew Abraham**[2], **Sergios Gatidis**[1]

**Akshay S Chaudhari**[1], **Curtis Langlotz**[1]

[1]Stanford University    [2]Stability AI

{zhihongc,mvarma2,jbdel,paschali,akshaysc,langlotz}@stanford.edu

## Abstract

Chest X-rays (CXRs) are the most frequently performed imaging test in clinical practice. Recent advances in the development of vision-language foundation models (FMs) give rise to the possibility of performing automated CXR interpretation. In this work, we present (i) *CheXinstruct* - a large-scale instruction-tuning dataset curated from 28 publicly-available datasets; (ii) *CheXagent* - an instruction-tuned FM capable of analyzing and summarizing CXRs; and (iii) *CheXbench* - a novel benchmark designed to systematically evaluate FMs across 8 clinically-relevant CXR interpretation tasks. Extensive quantitative evaluations and qualitative reviews with five expert radiologists demonstrate that CheXagent outperforms previously-developed general- and medical-domain FMs on CheXbench tasks by up to 97.5%.[1]

## Introduction

Foundation models (FMs) have recently emerged as a powerful class of models capable of performing a diverse range of reasoning and comprehension tasks (Bommasani et al. 2021). In this work, we present the following three components, also summarized in Fig. 1, to help create capable and robust FMs for chest X-ray (CXR) interpretation:

1. *CheXinstruct* is an instruction-tuning dataset with 6M instruction-image-answer triplets designed to improve the ability of FMs to interpret CXRs. We collect instructions from 34 tasks and 65 unique datasets, spanning categories including coarse- and fine-grained image understanding, question answering, and text generation.

2. *CheXagent* is an instruction-tuned foundation model with 8B parameters capable of analyzing images, understanding text, and generating responses. Our methodology for developing CheXagent includes training (1) a clinical LLM capable of understanding radiology reports, (2) a vision encoder capable of reading CXRs, and (3) a network to bridge the vision and language modalities. We then perform instruction-tuning using data from CheXinstruct.

3. *CheXbench* is a novel benchmark designed to rigorously evaluate FMs across two evaluation axes: image perception and textual understanding. We introduce 8 tasks

*Equal contributions.

[1]Our project is at https://stanford-aimi.github.io/chexagent.html.

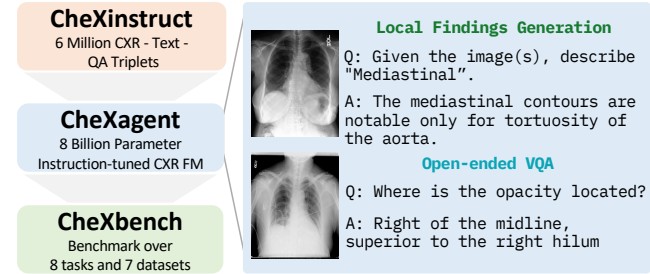

Figure 1: Overview of the proposed pipeline: CheXinstruct is a curation of datasets for instruction-tuning across various CXR tasks, CheXagent is our clinical FM for CXR interpretation, and CheXbench is our comprehensive FM evaluation benchmark. Two example CXR interpretation tasks include local findings generation and open-ended visual question answering (VQA).

across 7 CXR datasets, and we evaluate performance using close-ended multiple-choice predictions as well as open-ended text generation.

We use CheXbench to compare CheXagent with six previous FMs from both general and medical domains. We further provide an evaluation of potential model bias and highlight performance disparities across demographic factors of sex, race and age to improve model transparency.

## Data: CheXinstruct

CheXinstruct seeks to cover a broad range of tasks to support training CXR FMs. These tasks can either (i) improve the abilities of FMs to understand CXRs or (ii) improve clinical decision making. This dataset is comprised of five categories of tasks, each categorized by their specific capabilities:

- Coarse-grained Image Understanding, which defines the overall understanding of CXRs, e.g., view classification (Johnson et al. 2019), and disease classification (Wang et al. 2017; Irvin et al. 2019; Reis et al. 2022; Pavlova et al. 2022; Holste et al. 2023; Bannur et al. 2023; Jaeger et al. 2014; Bustos et al. 2020; Shih et al. 2019).

- Fine-grained Image Understanding, which defines the localized understanding of CXRs, e.g., abnormality detection (Nguyen et al. 2022; Pham, Tran, and Nguyen 2022),

Table 1: Comparison between CheXagent and general domain and medical domain FMs on CheXbench. For image perception tasks, we report accuracy; For Findings Generation and Findings Summarization, we report RadGraph Score and Rouge-L, respectively.

| Task | Dataset | General-domain FMs | | | Medical-domain FMs | | | CheXagent (Ours) |
|------|---------|--------|-------------|--------|-------------|-------|----------|------------------|
| | | BLIP-2 | InstructBLIP | XrayGPT | MedFlamingo | RadFM | LLaVA-Med | |
| View Classification | MIMIC-CXR | 28.8 | 25.3 | 24.0 | 25.0 | 28.5 | 23.8 | 97.5 |
| | CheXpert | 38.0 | 34.0 | 33.0 | 39.0 | 37.0 | 30.0 | 96.7 |
| Binary Disease Classification | SIIM | 53.0 | 54.0 | 50.0 | 50.0 | 50.0 | 49.0 | 64.0 |
| | RSNA | 50.0 | 60.0 | 50.0 | 50.0 | 50.0 | 44.0 | 81.0 |
| | CheXpert | 51.5 | 53.2 | 51.5 | 48.5 | 55.8 | 47.6 | 76.0 |
| Single Disease Identification | OpenI | 40.2 | 40.2 | 45.4 | 39.0 | 42.2 | 43.8 | 47.0 |
| | MIMIC-CXR | 25.6 | 22.6 | 24.1 | 25.6 | 27.2 | 26.7 | 30.3 |
| | CheXpert | 21.3 | 19.5 | 23.7 | 26.0 | 26.6 | 26.0 | 29.6 |
| Multi Disease Identification | OpenI | 48.5 | 54.4 | 57.7 | 46.1 | 52.8 | 53.9 | 55.6 |
| | MIMIC-CXR | 30.0 | 25.3 | 39.0 | 14.7 | 22.3 | 28.7 | 55.3 |
| | CheXpert | 4.3 | 6.1 | 3.9 | 7.1 | 23.6 | 2.1 | 52.1 |
| Visual Question Answering | Rad-Restruct | 41.2 | 42.4 | 38.6 | 45.5 | 48.5 | 34.9 | 57.1 |
| | SLAKE | 74.3 | 86.4 | 52.4 | 64.8 | 85.0 | 55.5 | 78.1 |
| Image-Text Reasoning | OpenI | 47.9 | 52.6 | 52.4 | 54.7 | 54.0 | 45.8 | 59.0 |
| Findings Section Generation | CheXpert | - | - | 9.0 | 1.7 | 5.1 | 4.2 | 14.6 |
| Findings Summarization | MIMIC-CXR | - | - | - | - | - | - | 40.3 |

abnormality grounding (Boecking et al. 2022), and foreign object detection (Xue et al. 2015).

- Question Answering, which defines the ability to respond to a question, e.g., close-ended and open-ended visual question answering (VQA) (Zhang et al. 2023; Pellegrini et al. 2023; Ben Abacha et al. 2019; Bae et al. 2023), difference VQA (Hu et al. 2023), and text QA.
- Text Generation, which defines the ability to generate radiology report sections, including a description of the findings (Demner-Fushman et al. 2012; Vayá et al. 2020; Pelka et al. 2018), impression generation (Feng et al. 2021), findings summarization (Chen et al. 2023), and local findings generation (Johnson et al. 2019).
- Miscellaneous: This category defines the miscellaneous abilities that are critical for a CXR FM, e.g., report evaluation (Yu et al. 2023; Miura et al. 2021), and natural language explanation (Kayser et al. 2022).

## Model: CheXagent

The aim of CheXagent is a model that can "see" images $x_I$ and/or "read" text $x_T$ and generate "responses" $y$. Our training process for CheXagent involves the following stages: **Stage 0: Train a clinical LLM**: Our starting point is Mistral-7B-v0.1 (Jiang et al. 2023) due to its proven robust reasoning abilities in diverse benchmarks. To infuse the model with comprehensive medical and clinical knowledge, we utilize five distinct text sources for training: (i) PMC articles, (ii) MIMIC-IV, and (iii) Wikipedia. **Stage 1: Train a vision encoder for CXR**: Our model architecture reflects that of (Li et al. 2023c). For training purposes, we utilize datasets comprising image-text pairs, specifically from MIMIC-CXR, PadChest, and BIMCV-COVID-19. **Stage 2: Train a vision-language bridger**: Following the training of the clinical LLM and the CXR vision encoder, we focus on developing a bridger model, $\mathcal{M}_b$. This model is designed to map visual data to the corresponding language (semantic) space. For training $\mathcal{M}_b$, we employ the same datasets as in Stage 1. **Stage 3: Instruction tuning**: Upon completing Stage 2, The model is trained on CheXinstruct, taking into account two primary factors: (i) reserving certain task-dataset pairs

exclusively for evaluation purposes, and (ii) determining optimal dataset ratios to ensure balanced training across different capabilities.

## Evaluation: CheXbench

CheXbench is structured with two evaluation axes, crafted to assess crucial aspects of CXR interpretation: image perception and textual understanding. For the former, we introduce 6 tasks across 7 datasets: View Classification, Binary Disease Classification, Single Disease Identification, Multi-Disease Identification, Visual-Question-Answering, and Image-Text Reasoning; For the latter, we introduce 2 tasks: Findings Section Generation and Findings Summarization.

In our study, we employ CheXbench to compare CheXagent against two general-domain instruction-tuned FMs, InstructBLIP and BLIP2, which achieve state-of-the-art performance in previous research (Li et al. 2023a). Additionally, we compare CheXagent with four medical FMs: XrayGPT, MedFlamingo, RadFM, and LLaVA-Med (Thawkar et al. 2023; Moor et al. 2023; Li et al. 2023b; Wu et al. 2023). This comparison aims to provide a comprehensive understanding of CheXagent's performance in relation to both general and medical-specific models.

Table 1 provides results on CheXbench. For image perception tasks, CheXagent demonstrates superior performance across image perception tasks, achieving an average improvement of 97.5% over general-domain FMs and an average improvement of 55.7% over medical FMs; For text understanding tasks, CheXagent outperforms all medical FMs on CheXpert on findings section generation and also achieve promising performance on findings summarization.

## Conclusion

In this work, we design a complete scheme for training CXR FMs by introducing CheXisntruct, CheXagent, and CheXbench. Experimental results demonstrate the effectiveness of this scheme.

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
