# OpenReview forum: "CheXagent: Towards a Foundation Model for Chest X-Ray Interpretation"
_AAAI.org/2024/Spring_Symposium_Series/Clinical_FMs — AAAI 2024 SSS on Clinical FMs_

### Official Review · Reviewer_oVSn · 2024-02-20
**Chest X-Ray foundation model**

**Rating:** 9
**Confidence:** 4

**Review:**

In this work the authors bundle a pre-training dataset consisting of several open source cohorts, a vision-language model trained on said dataset, and a suite of benchmarking tasks to compare performance of foundation models in the space of chest X-rays interpretation.

The work is impressive and might have benefited from a longer write-up as with the 2 page constraint it is challenging to thoroughly describe all aspects of the work. I would have liked to see a less succinct explanation of the model and the datasets, in particular how the splitting between training and testing was done. Additionally, a section regarding limitations of the current work and ways into clinical application would have been appreciated.

---

### Official Review · Reviewer_kB8R · 2024-02-22
**A Foundation Model for Chest X-Ray Interpretation**

**Rating:** 7
**Confidence:** 4

**Review:**

**Summary**:

The authors present a foundation model for chest x-ray interpretation with three novel components, they present an instruction-tuning dataset, a foundation model trained on this dataset, and a benchmarking tool to evaluate FMs across different CXR datasets. The authors showcase the performance of their FM across multiple tasks, such as Image Perception, Question Answering, and Text Generation. The authors compare their FM against other General and Medical-domain specific FMs across 8 tasks and 7 CXR datasets and indicate a general superiority of the performance of their models across the different tasks.

**Pros**:

- *CheXagent's* performance looks impressive across the different tasks. They are able to demonstrate its utility as a Foundation Model by displaying generally superior performance across 7 tasks that are relevant to an FM in this domain.
- The authors comprehensively describe the steps they took to infuse the underlying LLM with medical and clinical knowledge. The overall architecture of *CheXagent* also appears convincing and matches intuition on their good performance across the different tasks.
- The authors have created a benchmark for evaluating FMs for Chest X-rays using their *CheXbench* benchmark. This is important for reproducibility and extending the evaluation across newer Chest X-rays datasets and Medical FMs.

**Cons**:

- The authors mention they provided an evaluation of the disparities of the mode's performance across demographics, and important considerations for FMs when they envision their FMs being adopted into clinical practice by radiologists.
- I would've liked the authors to provide a list of diseases they are evaluating the models on along with their distributions. It would have been interesting to provide some examples of the model's performance in the appendix.
- I would've liked the authors to make a comment or two on interpretability, and how they think they could extend their work to incorporate measures of explainability and trustworthiness, which is fundamental to the adoption of AI models into clinical practice. Frequently, we run into the issue of developing "black-box" models in the field with not a lot of effort made into explaining how the model makes its decisions.

**Quality**:

The overall quality of the paper is good.

**Originality**:

Even though it appears that some work has been done on building summarization models for Chest X-Rays using Multimodal LLMs like GPT, I believe the authors have done a good job highlighting the novelty of their work in compounding an extensive X-ray dataset, developing a novel FM specific to Chest X-ray and evaluating it across a variety of tasks.

**Significance**:

I believe this can be a significant model in the field of radiology if the authors are able to extend this work to take it out of its current "black-box" setting and integrate some form of explainability or generalization, and comprehensively evaluate how this model's performance differs across different demographics.

**Miscellaneous Comments**:

- Have the authors considered incorporating the Noisy CXR dataset into CheXinstruct? The dataset presents the unique dimension of capturing label noise, and it would be interesting to observe how CheXagent performs under such a scenario.

---

### Official Review · Reviewer_jwjZ · 2024-02-22
**Paper is fine, but is evaluation is lacking upper bounds in terms of expert models.**

**Rating:** 6
**Confidence:** 5

**Review:**

The paper presents a foundation model based on a custom dataset called CheXinstruct. The authors train a BLIP-2 model on the proposed dataset and evaluate on several downstream tasks. For the metrics the authors choose accuracy as stated in table 1, which seems to be ill-chosen in the chest x-ray domain since there is a massive class imbalance. Also in the proposed evaluation benchmark a comparison from the generalist BLIP-2 model based on mistral 7B  to expert models specific to the individual tasks as a quasi upper bound is missing. Especially for the classification tasks this might provide some better perspective on the difficulty of the task. Similarly, a simple baseline might be beneficial to get a grasp of the lower bound of each task.
Overall there are some serious concerns regarding the evaluation protocoll of this paper which might be resolved by the use of better suited metrics and comparison models.

---

### Official Review · Reviewer_VAKt · 2024-02-23
**Foundational model for chest X-ray analysis, including a specialized dataset**

**Rating:** 9
**Confidence:** 3

**Review:**

This work lays the groundwork for the creation of a Foundation Model (FD) on the domain of Chest X-Ray (CXR) images. It proposes a set of datasets (CheXinstruct) containing 5 task categories related to CXR, a FD trained on these datasets (CheXagent) and an evaluation benchmark (CheXbench), in which the model achieves outstanding results. This work is well presented, clearly expressed and is especially relevant with regards to the topic of Foundational Models for medicine. Although it builds on the contributions of many other works (which is properly cited in the text), it is an important contribution nonetheless and will allow further research and advances in the topic. For these reasons, I consider to accept this paper for the symposium.

I would like to include some comments for further clarification or correction of the work.

- When listing the tasks in the construction of the dataset, some tasks might not be familiar for all readers and a short description would be beneficial, e.g., view classification.
- In the model section, “To infuse the model […], we utilize **five** distinct text sources for training”, but then only three are mentioned. Either a clarification or a correction is needed here.
- For training stage 2, training a vision-language bridger, a reference is needed to clarify how this process takes place.
- Regarding the evaluation table, no confidence intervals are given, which makes it harder to compare between models with similar metrics, particularly for single- and multi- disease identification.
- No results are given for any other method in the Findings Summarization task; if it is in the author’s power to do so, it would be beneficial to have results for these methods.
- Regarding the outstanding results in terms of View Classification, a comment on the reasons for this dramatic increase are missing.